# Association of Muscle Strength with Non-Alcoholic Fatty Liver Disease in Korean Adults

**DOI:** 10.3390/ijerph19031675

**Published:** 2022-02-01

**Authors:** Sung-Bum Lee, Yu-Jin Kwon, Dong-Hyuk Jung, Jong-Koo Kim

**Affiliations:** 1Department of Medicine, Graduate School, Yonsei University Wonju College of Medicine, Wonju 26426, Korea; dolsoui@yuhs.ac; 2Department of Family Medicine, Yongin Severance Hospital, Yonsei University College of Medicine, Yongin 16995, Korea; digda3@yuhs.ac (Y.-J.K.); balsan2@yuhs.ac (D.-H.J.); 3Department of Family Medicine, Yonsei University Wonju College of Medicine, Wonju 26426, Korea; 4Research Group for Global Health and Medical Technology Development, Yonsei University Wonju College of Medicine, Wonju 26426, Korea

**Keywords:** handgrip strength, sarcopenia, NAFLD

## Abstract

Sarcopenia is known to be associated with non-alcoholic fatty liver disease (NAFLD). However, few studies have revealed the association between muscle strength and prevalence of NAFLD. We investigated the association by using relative handgrip strength in a nationwide cross-sectional survey. The participants were recruited from the Korean National Health and Nutrition Examination Surveys (KNHANES). A total of 27,531 subjects from the KNHANES were selected in our study. We used normalized handgrip strength, which is called relative handgrip strength. The index was defined as handgrip strength divided by BMI. These subjects were divided into quartile groups according to relative handgrip strength. NAFLD was defined as hepatic steatosis index >36. Multinomial logistic regression was analysed to investigate the association between relative handgrip strength with prevalence of NAFLD. The mean age of study population was 45.8 ± 0.3 in men, and 48.3 ± 0.2 in women. The proportion of males was 37.5%. In multiple linear regression, relative handgrip strength was inversely associated with HSI index (Standardized β = −0.70; standard error (SE), 0.08; *p* < 0.001 in men, Standardized β = −0.94; standard error (SE), 0.07; *p* < 0.001 in women). According to the logistic regression model, the prevalence of NAFLD decreased with quartile 4 groups in relative handgrip strength, compared with quartile 1 groups (OR 0.42 [0.32–0.55] in men; OR 0.30 [0.22–0.40] in women). Relative handgrip strength, used as a biomarker of sarcopenia, is independently inversely associated with NAFLD.

## 1. Introduction

Sarcopenia, a condition of decreasing skeletal muscle mass and strength, is a major health concern worldwide [1]. It has attracted research interest in recent decades due to its comorbidities. It increases the risk for metabolic disease, diabetes, and mortality [2,3]. Furthermore, the prevalence of sarcopenia varies worldwide from 10% to 40% after adjusting for different criteria of sarcopenia [4]. This is expected to gradually rise during the next 30 years [5].

Previous studies have demonstrated an independent relationship between skeletal muscle mass and severity of non-alcoholic fatty liver disease (NAFLD) [6,7]. However, the association between sarcopenia and NAFLD has been evaluated from muscle mass rather than muscle strength. Recently, many studies have suggested that muscle strength is a more precise indicator than muscle mass for predicting health outcomes [8]. Nevertheless, few studies on the association between muscle strength and NAFLD have been reported [9,10].

Handgrip strength is an inexpensive and quick method that is widely used to represent muscle strength because of its high correlation of whole-body strength including leg strength [11,12]. Furthermore, handgrip strength is an attractive tool to stratify the risk of many diseases. Therefore, there are many articles on handgrip strength and its association with health problems. Many studies have shown that handgrip strength is associated with metabolic disease, diabetes, and hypertension [13,14,15]. With the emerging importance of handgrip strength, recent studies have also used relative handgrip strength (absolute handgrip strength/BMI) [16,17]. A previous study demonstrated that relative handgrip strength is associated with metabolic diseases [18]. However, there are few studies that examine relative handgrip strength and NAFLD [10,19].

Therefore, we aimed to examine the association between relative handgrip strength and prevalence of NAFLD in adults through an analysis of a nationwide cross-sectional database over 5 years.

## 2. Materials and Methods

### 2.1. Study Population

In this cross-sectional study, we analysed the Korean National Health and Nutrition Examination Surveys (KNHANES) data from 2014 to 2018, retrospectively. The KNHANES is a national surveillance system to assess the health and nutritional status of Koreans and performed by the Korea Centres for Disease Control and Prevention in the Ministry of Health and Welfare, which has previously reported with the details [6].

Figure 1 shows the schematic process of participant selection. Among a total of the 32,704 participants, we excluded subjects who met the following criteria: (1) age ≤ 19 years; (2) absence of handgrip strength data; (3) alcohol consumption per week >140 g for men and >70 g for women [20]; (4) positive for hepatitis B surface antigen or hepatitis c antibody [21]. After inspecting the exclusion criteria, 19,852 subjects were included in our analysis. All participants signed an informed consent form.

### 2.2. Anthropometric and Laboratory Measurements

Anthropometric data, health-related factors, and blood markers were obtained for all subjects. Data included age, waist circumference (WC), systolic blood pressure (SBP) and diastolic blood pressure (DBP). WC was measured by using a flexible tape (Seca 220; Seca), and examined based on the midpoint the lowest margin of the rib and the uppermost border of the iliac crest during expiration [22,23]. All BP were measured using a mercury sphygmomanometer after the participants had rested for 5 min in a sitting position (Baumanometer Wall Unit 33(0850)). All BP examinations were conducted on the right arm three times using the same instruments at 30-s intervals with a cuff appropriate for arm circumference [24]. Health-related factors included alcohol uptake, smoking history, regular exercise, hypertension, and diabetes. Alcohol uptake was defined as those who drank alcohol at least once a week; alcohol consumption per week >140 g for men and >70 g for women. Smokers were defined as current smokers, ex-smokers based on questionnaires (ex-smokers were defined based on the response “I have smoked more than 5 packs of cigarettes but do not smoke any more”). Regular exercise was defined as performing either moderate or vigorous physical activity more than three times per week. The Global Physical Activity Questionnaire (GPAQ) was standardized to measure the level of physical activity of people [25]. Hypertension was diagnosed as systolic blood pressure ≥140 mmHg, diastolic blood pressure ≥90 mmHg, or taking antihypertensive medications [26]. Diabetes was defined as 8-h fasting glucose levels ≥126 mg/dL, glycated haemoglobin (HbA1c) level ≥6.5%, taking antidiabetic medications including insulin, or diagnosis by a physician [27]. HbA1c levels were decided by performance liquid chromatography using an automated HGLC-723G7 analyzer (Tosoh Corporation, Tokyo, Japan) [28]. The medication history and diagnosis by a physician have been obtained based on questionnaires in the data. Fasting plasma blood glucose, total cholesterol (TC), triglyceride (TG), alanine aminotransferase (ALT), and aspartate aminotransferase (AST) were measured using the Hitachi Automatic Analyzer 7600-210 (Hitachi, Tokyo, Japan) [28].

### 2.3. Assessment of Handgrip Strength

Handgrip strength was measured three times using a digital grip strength dynamometer (TKK5401; Takei Scientific Instruments Co, Ltd., Tokyo, Japan) [29]. It was measured with the subjects in a standing position and with the arms in full extension. The participants were instructed to squeeze the dynamometer with as much force as possible, for at least three seconds, three times with each hand alternatively. A rest interval of one minute was given between each trial. Absolute handgrip strength was defined as the summation of the maximum value from each hand and was expressed in kilograms. Handgrip strength is known to be correlated with body size (BMI, weight, height). There are several studies to reduce the effects of body size on handgrip strength, such as hand grip force divided by weight, or height^2^, or BMI [30,31]. Among these indexes, we used relative handgrip strength that is calculated as the absolute handgrip strength divided by BMI because it is previously used as an indicator for muscle strength [32]. The relative handgrip strengths were divided into sex-specific quartiles.

### 2.4. Definition of Non-Alcoholic Fatty Liver Disease

Non-alcoholic fatty liver disease (NAFLD) was diagnosed by a well-validated fatty liver predictor, the hepatic steatosis index (HSI), calculated as 8 * alanine aminotransferase (ALT)/aspartate aminotransferase (AST) + BMI (+2 if diabetes; +2 if female). NAFLD was defined as HSI > 36 [33]. The sensitivity and specificity of HIS for prediction of NAFLD are 86% and 66%, respectively, in the Korean population [33].

### 2.5. Statistical Analysis

All the variables of the study population were analysed by independent t-tests and analysis of variance (ANOVA) tests for continuous variables and chi-square tests for categorical variables. Multiple linear regression was conducted to investigate the association of relative handgrip strength and HSI after adjusting for age, WC, regular exercise, smoking history, fasting plasma glucose, TC, TG, and SBP. Levels of relative handgrip strength were categorized into four groups according to quartile value: Q1, ≤2.8; Q2, 2.81–3.3; Q3, 3.31–3.7; Q4, >3.7 in men, and Q1, ≤1.7; Q2, 1.71–2.0; Q3, 2.01–2.4; Q4, >2.4 in women. The weakest relative handgrip strength group (Q1) was considered as the reference group. Multinomial logistic regression analysis was performed to calculate the odds ratios (ORs) and 95% confidence intervals (95% CIs) of NAFLD for relative handgrip strength after adjusting for the confounding factors across relative handgrip strength quartiles. Weight values were applied to all variables. The values were made for sample participants to represent the Korean population by accounting for the non-response, complex survey design and post-stratification [34]. *p*-values < 0.05 were defined as statistically significant. Statistical analyses were performed using IBM SPSS Statistics (Version 22.0, IBM Corp., Armonk, NY, USA).

## 3. Results

Table 1 shows the baseline characteristics of the 19,852 participants (7443 men, and 12,409 women) according to relative handgrip strength quartile. The mean age of the men and women was 45.8 ± 0.3 and 48.3 ± 0.2 years respectively, with 37.5% of the sample being male. The mean HSI was higher in men than women (33.3 in men, 32.0 in women). The mean value of age, WC, fasting plasma glucose, SBP, hypertension, diabetes, and HSI decreased with increasing relative handgrip strength quartile in both sexes. Total cholesterol and TG were increased more in Q2 than Q1 in men. However, these variables decreased gradually from Q2 to Q4, which was a similar trend to total cholesterol in women. On the other hand, the mean value of regular exercise increased in both sex along with quartile.

NAFLD prevalence was reduced with increasing relative handgrip strength quartile in both sex (Figure 2). This shows that a dose-response association was observed between relative handgrip strength and NAFLD.

Table 2 presents the results of the linear regression analyses to investigate the associations between relative handgrip strength and HSI. Higher relative handgrip strength was inversely associated with HSI in men.

Table 3 indicates the results of the logistic regression analyses induced to assess the relationship between relative handgrip strength quartile and NAFLD. We set the first quartile of relative handgrip strength as the reference group [13]. Compared with the reference group (Q1), the ORs for NAFLD of subjects in Q4 were 0.16 (95% CI, 0.13–0.20) in men and 0.05 (95% CI, 0.04–0.07) in women, when unadjusted. After adjusted for age (Model 1), the ORs for NAFLD in Q4 were 0.10 (0.08–0.13) in men and 0.06 (0.04–0.07) in women. After further adjusting model 1 for WC, regular exercise, and smoking status, the ORs (95% CI) in Q4 were 0.43 (0.33–0.56) in men and 0.30 (0.23–0.40) in women, which were slightly attenuated; however, they still maintained statistical significance. After further adjusting model 2 for fasting plasma glucose, total cholesterol, TG, and SBP, the ORs (95% CI) in Q4 were 0.42 and 0.30 in men and women, respectively, which remained statistically significant.

## 4. Discussion

In this large, cross-sectional population study conducted over 5 years, relative handgrip strength was independently inversely associated with prevalence of NAFLD in adults without excessive alcohol uptake. Handgrip strength was an independent risk factor for NAFLD regardless of age, sex, WC, BMI, regular exercise, smoking history, fasting glucose, total cholesterol, TG, and SBP. Furthermore, there was a dose-response association between handgrip strength and the prevalence of NAFLD.

Handgrip strength is a well-known prognostic indicator for hypertension and metabolic syndrome [13,15]. Moreover, many studies have shown that handgrip strength is a useful biomarker of cognitive function like Alzhemier`s disease and mild cognitive impairment [35]. However, there are few studies that show the relationship between handgrip strength and NAFLD. We demonstrated the relationship between handgrip strength and fatty liver in a number of subjects extracted from the 32,704 participants over 5 years; our results are independent of alcohol history by excluding subjects with excessive alcohol uptake.

Low muscle strength is used as a principal determinant of sarcopenia over low muscle mass because muscle strength is more important to predict fracture, falling, and all-cause mortality [14,36]. Furthermore, handgrip strength is recommended as a substitute measurement of muscle strength due to its quick and easy-to-obtain examination according to the European Working Group on Sarcopenia in Older People [11]. Therefore, handgrip strength is usually applied as a diagnostic approach for sarcopenia [37].

Several mechanisms may affect an inverse relationship between sarcopenia and NAFLD [38]. Crucial mechanisms involved in the interaction of sarcopenia and NAFLD are based on effects of insulin resistance (IR), oxidative stress, and chronic inflammation [39,40]. Insulin plays an important role in glucose metabolism, and liver and skeletal muscle are target organs of insulin. IR is a pathological condition in which cells fail to respond to insulin [38]. IR may cause fat tissue infiltration in skeletal muscle accompanied by circulating free-fatty acid (FFA) from excessive body fat [41]. Moreover, IR of skeletal muscle results in reduction of protein synthesis, which leads to muscle loss. Therefore, IR contributes to sarcopenia. On the other hand, sarcopenia can aggravate IR. Skeletal muscles have a primary role in glucose homeostasis by expression of insulin-dependent transporter GLUT-4. If sensitivity to insulin decreases, uptake of glucose is impaired, and insulin promotes glycogen synthesis [41]. Accordingly, conversion of glucose to triacylglycerol is increased in the liver, which causes development of fatty liver. This pathogenesis can explain a causal link between IR and fatty liver. Besides, this process is connected more to skeletal IR than to hepatic IR, which can be independent of alcohol intake, because alcohol consumption is related to hepatic IR by inhibiting fatty acid oxidation through suppression of PPAR-α in hepatocytes [42,43]. Consequently, IR is a pivotal mechanism involved in handgrip strength and NAFLD. Oxidative stress and inflammation are mutually involved in IR, sarcopenia, and NAFLD. TNF-α stimulates production of reactive oxygen species, leading to oxidative stress and mitochondrial dysfunction. Moreover, it also inactivates the AMP-activated protein kinase pathway, which is associated with NAFLD development. IL-6, a proinflammatory cytokine, also plays a pivotal role in inflammation and NAFLD development. Both cytokines are inversely associated with sarcopenia. Previous studies have shown an association of increased cytokines (e.g., TNF- α, IL-6, hs-CRP) with sarcopenia and progression of NAFLD [40,44,45]. For instance, subjects with sarcopenia also had higher levels of hs-CRP compared with participants without sarcopenia [45]. These data suggest that inflammation can be involved in the pathogenetic mechanisms of sarcopenia and NAFLD.

In spite of our recruitment of a large number of subjects, there are several limitations to our study. First, as there are no muscle mass data in the current KNHANES data, we cannot confirm if the relationship between handgrip strength and NAFLD is independent of muscle mass. Nevertheless, we used the data because handgrip strength more exactly represents sarcopenia than muscle mass [14]. Recently, sarcopenia is defined as muscle failure, with low muscle strength overtaking muscle mass, which shows that muscle strength is a primary indicator of sarcopenia over muscle mass [46,47]. Thereby, handgrip strength is used as a mandatory measurement for the diagnosis of sarcopenia in Europe [11]. Second, we defined NAFLD by HSI index, not imaging study and histology. In general, liver biopsy is the gold standard method to diagnose NAFLD, and imaging studies such as ultrasonography or MRI can be alternative diagnostic tools. However, these methods were not available in the large population-based study because of their invasive procedure and cost. In this sense, there are previous studies that have used HSI to diagnose NAFLD [48,49]. Furthermore, because it is a cross-sectional study, we cannot explain the causality of handgrip strength and NAFLD. Finally, an appropriate indicator to completely eliminate the effect of body size on handgrip strength has not been determined. Although relative handgrip strength was been used to reduce the effect of body size, even relative handgrip strength cannot completely eliminate the effect of body size [30]. In the future, it seems that additional research is needed on muscle strength-related indicators independent of body size, especially in the Korean population.

With these limitations, compared with previous studies, this study used representative national data, and included a relatively large number of subjects of more than 30,000 over 5 years. We suggested that muscle strength is associated with the prevalence of NAFLD, which is a more useful marker than muscle mass by using handgrip strength.

## 5. Conclusions

In conclusion, muscle strength is independently inversely related to the prevalence of NAFLD. Relative handgrip strength can be used as a substitute measurement of muscle strength. Timely detection of handgrip strength is crucial to predict the prevalence of NAFLD.

## Figures and Tables

**Figure 1 ijerph-19-01675-f001:**
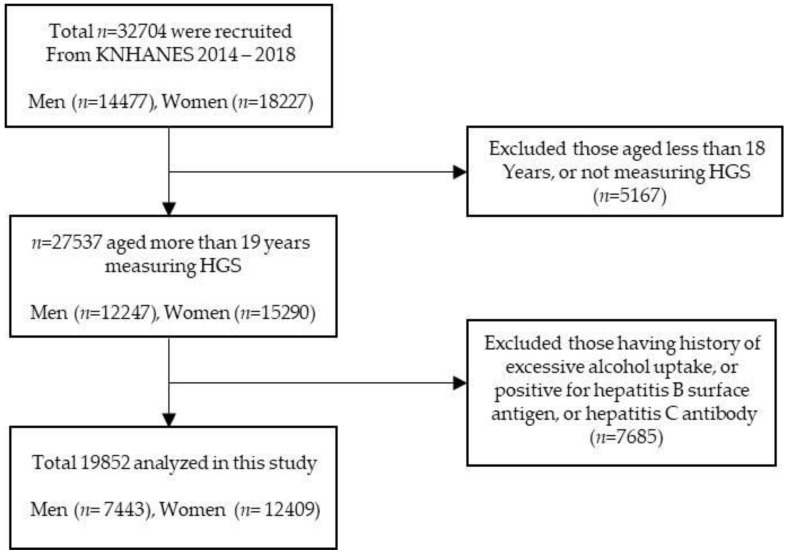
Flow chart of the study population selection.

**Figure 2 ijerph-19-01675-f002:**
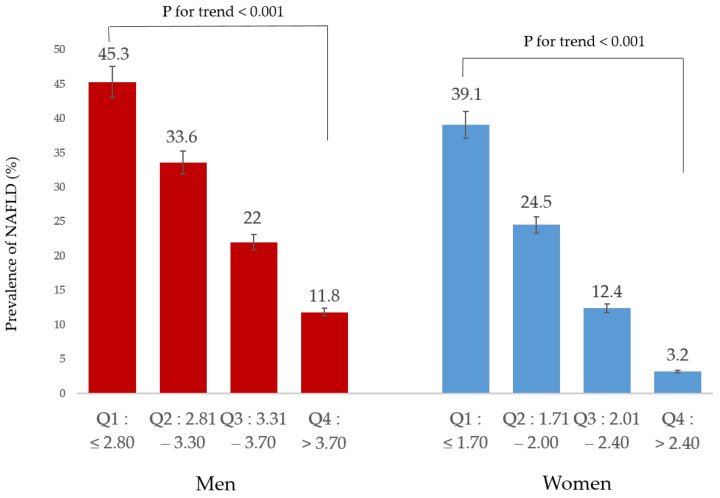
Prevalence of NAFLD according to relative handgrip strength.

**Table 1 ijerph-19-01675-t001:** Baseline characteristics according to RGS quartile.

	Men	Q_1_	Q_2_	Q_3_	Q_4_	*p*-Value
		≤2.80	2.81–3.30	3.31–3.70	>3.70	
N	7443	2259	2069	1451	1664	
Age (years)	45.8 ± 0.3	52.8 ± 0.6	47.5 ± 0.5	44.0 ± 0.5	38.8 ± 0.4	<0.001
WC (cm)	85.3 ± 0.1	90.4 ± 0.3	86.8 ± 0.2	84.1 ± 0.2	80.0 ± 0.2	<0.001
Fasting glucose (mg/dl)	101.0 ± 0.3	106.4 ± 0.7	103.2 ± 0.6	99.1 ± 0.7	95.4 ± 0.4	<0.001
Total cholesterol (mg/dl)	188.9 ± 0.5	187.6 ± 1.0	191.2 ± 1.0	191.0 ± 1.0	186.8 ± 1.0	0.001
Triglyceride (mg/dl)	147.2 ± 1.5	155.4 ± 2.8	158.5 ± 3.2	145.7 ± 3.4	128.8 ± 2.8	<0.001
Systolic BP (mmHg)	118.4 ± 0.2	122.3 ± 0.4	119.3 ± 0.3	117.6 ± 0.4	114.4 ± 0.3	<0.001
Smoking status, *n* (%)						<0.001
Never smoker	2125 (31.7)	618 (32.1)	590 (32.1)	429 (33.0)	488 (31.2)	
Ex-smoker	3127 (36.6)	1104 (41.8)	899 (38.0)	550 (34.2)	574 (31.9)	
Current smoker	2185 (31.7)	532 (26.0)	580 (29.9)	471 (32.8)	602 (36.9)	
Regular exercise, *n* (%)	2168 (33.8)	475 (25.6)	589 (32.3)	484 (36.7)	620 (40.8)	<0.001
Hypertension, *n* (%)	2393 (26.3)	1087 (41.0)	705 (28.4)	345 (21.1)	256 (13.7)	<0.001
Diabetes, *n* (%)^b^	1145 (12.1)	561 (21.1)	356 (14.4)	141 (8.3)	87 (4.3)	<0.001
HSI	33.3 ± 0.1	36.0 ± 0.2	34.3 ± 0.2	32.6 ± 0.2	30.5 ± 0.1	<0.001
	**Women**	**Q1**	**Q2**	**Q3**	**Q4**	** *p* ** **-Value**
		**≤1.70**	**1.71–2.00**	**2.01–2.40**	**>2.40**	
N	12409	3637	2638	3440	2694	
Age (years)	48.3 ± 0.2	58.3 ± 0.4	50.0 ± 0.4	44.2 ± 0.3	39.5 ± 0.3	<0.001
WC (cm)	78.2 ± 0.1	84.7 ± 0.2	80.2 ± 0.2	76.0 ± 0.2	71.8 ± 0.2	<0.001
Fasting glucose (mg/dl)	97.4 ± 0.2	104.5 ± 0.5	98.4 ± 0.5	94.7 ± 0.4	91.8 ± 0.3	<0.001
Total cholesterol (mg/dl)	192.1 ± 0.4	194.9 ± 0.8	196.0 ± 0.8	192.3 ± 0.7	185.8 ± 0.7	<0.001
Triglyceride (mg/dl)	112.0 ± 0.9	136.1 ± 1.8	117.7 ± 1.9	104.8 ± 1.4	89.1 ± 1.2	<0.001
Systolic BP (mmHg)	114.8 ± 0.2	122.0 ± 0.4	116.6 ± 0.4	111.6 ± 0.3	108.5 ± 0.3	<0.001
Smoking status, *n* (%)						0.108
Never smoker	11372 (90.9)	3369 (92.2)	2433 (91.4)	3145 (90.6)	2425 (89.8)	
Ex-smoker	593 (5.3)	143 (4.2)	122 (5.0)	172 (5.4)	156 (6.2)	
Current smoker	435 (3.9)	120 (3.6)	79 (3.6)	123 (4.0)	113 (4.0)	
Regular exercise, *n* (%)	2646 (24.1)	449 (14.2)	545 (22.6)	859 (27.3)	793 (31.6)	<0.001
Hypertension, *n* (%)	3485 (23.7)	1782 (43.3)	826 (27.0)	635 (15.3)	242 (7.8)	<0.001
Diabetes, *n* (%)^b^	1431 (10.2)	811 (21.2)	317 (11.4)	220 (5.8)	83 (2.8)	<0.001
HSI	32.0 ± 0.1	35.1 ± 0.1	33.2 ± 0.1	31.2 ± 0.1	28.9 ± 0.1	<0.001

RGS: relative handgrip strength, WC: waist circumference, HSI: hepatic steatosis index.

**Table 2 ijerph-19-01675-t002:** Results of multiple linear regression analysis to assess the independent relationships between RGS and HSI in Koreans.

		Men	
	Standardized β	SE	*p*-Value
Unadjusted	−2.93	0.13	<0.001
Model 1	−3.17	0.14	<0.001
Model 2	−0.67	0.08	<0.001
Model 3	−0.70	0.08	<0.001

		**Women**	
	**Standardized β**	**SE**	***p*-Value**
Unadjusted	−4.46	0.10	<0.001
Model 1	−4.14	0.12	<0.001
Model 2	−1.01	0.07	<0.001
Model 3	−0.94	0.07	<0.001

RGS: relative handgrip strength, HSI: hepatic steatosis index, SE: standard error. Model 1: adjusted for age. Model 2: adjusted for age, waist circumference, regular exercise, and smoking status. Model 3: adjusted for age, waist circumference, regular exercise, smoking status, fasting glucose, total cholesterol, TG, and systolic blood pressure.

**Table 3 ijerph-19-01675-t003:** Odds ratio and 95% confidence intervals for NAFLD according to RGS in Koreans with the use of HSI.

	Men				Women			
	Q_1_	Q_2_	Q_3_	Q_4_	Q_1_	Q_2_	Q_3_	Q_4_
	≤2.80	2.81–3.30	3.31–3.70	>3.70	≤1.70	1.71–2.00	2.01–2.40	>2.40
Unadjusted	1.00	0.61 (0.52–0.71)	0.34 (0.28–0.41)	0.16 (0.13–0.20)	1.00	0.51 (0.44–0.58)	0.22 (0.19–0.25)	0.05 (0.04–0.07)
Model 1	1.00	0.49 (0.41–0.58)	0.24 (0.20–0.30)	0.10 (0.08–0.13)	1.00	0.52 (0.45–0.59)	0.24 (0.20–0.27)	0.06 (0.04–0.07)
Model 2	1.00	0.81 (0.66–0.99)	0.55 (0.43–0.72)	0.43 (0.33–0.56)	1.00	0.81 (0.69–0.95)	0.62 (0.52–0.73)	0.30 (0.23–0.40)
Model 3	1.00	0.77 (0.63–0.94)	0.53 (0.41–0.70)	0.42 (0.32–0.55)	1.00	0.82 (0.69–0.97)	0.62 (0.52–0.74)	0.30 (0.22–0.40)

RGS: relative handgrip strength, HSI: hepatic steatosis index. Model 1: adjusted for age. Model 2: adjusted for age, waist circumference, regular exercise, smoking status. Model 3: adjusted for age, waist circumference, regular exercise, smoking status, fasting glucose, total cholesterol, TG, and systolic blood pressure.

## Data Availability

The data underlying this article will be shared upon reasonable request from the corresponding author.

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
