# Peer review of "Association of Muscle Strength with Non-Alcoholic Fatty Liver Disease in Korean Adults"

_ijerph, 2022, doi:10.3390/ijerph19031675_

Round 1
Reviewer 1 Report
Thank you for the opportunity to contribute the relevant content covered in the manuscript. The aim of the study was to investigate the association between muscle strength assessed by handgrip and non-alcoholic fatty liver disease in Korean adults. In an analysis conducted with 19,852 subjects, the authors identified that normalized muscle strength was inversely associated with NAFLD regardless of metabolic syndrome. However, some issues should be further explored in relation to how body size was considered in determining the results (since normalizing the absolute values of handgrip strength for a simple reason will only imply modifying the direction of the impact exerted by body size in this interrelation, but it does not eliminate its impact – which will possibly impact the determination of the results obtained). Further details can be seen below, according to each section:
ABSTRACT
Reviewer: Please add briefly the information of the investigated sample (age group, gender, place where the data were collected).
Reviewer: I suggest describing in this section how the handgrip strength was normalized and what body-related index was used for such a procedure.
Reviewer: Please describe the words in full before adding abbreviations.
Ex: For those who are reading the work for the first time, the acronyms NAFLD or HBV/HCV will have no meaning. Please add such information!
- “As handgrip strength increased, the age, waist circumference (WC), fasting glucose, systolic BP, hypertension, diabetes, and hepatic steatosis index (HSI) are significantl y decreased”.
Reviewer: I was unable to verify the relevance of including the aforementioned sentence in the “abstract” section of this study. In my opinion, such sentence can be deleted from this section.
Reviewer: As well as logistic regression values, I suggest adding association values referring to multiple linear regression analysis (HSI index).
- "Relative handgrip strength, used as a biomarker of sarcopenia, is independently negatively associated with NAFLD".
Reviewer: Given the nature of the investigated associations (cross-sectional design), I think it is not coherent to add vocabulary that denotes causality (e.g. negatively associated). I suggest modifying the term used (e.g., inversely associated).
- "The relationship between handgrip strength and NAFLD was independent of metabolic syndrome".
Reviewer: In my opinion, since the relevance of the interrelationship of NAFLD x metabolic syndrome was never described, adding in the concluding section that the results were identified independently of metabolic syndrome does not seem to make sense to me. f the authors choose to maintain such a sentence, they must substantiate why this finding is relevant (results independent of metabolic syndrome). Otherwise, I suggest deleting!
- “Furthermore, its association may be mechanically linked through insulin resistance and inflammation”.
Reviewer: Do not include justification for findings in the concluding section. I suggest deleting!
INTRODUCTION
- “Sarcopenia, a condition of decreasing skeletal muscle mass and strength, is a major health concern worldwide”.
Reviewer: Please reference such information.
- “Nevertheless, few studies on the association between muscle strength and NAFLD have been reported”.
Reviewer: Could you please cite two or three studies that support this interrelationship?
- “With the emerging importance of handgrip strength, recent studies have also used relative handgrip strength (absolute handgrip strength/BMI), which is an independent indicator for BMI.”
Reviewer: Please add references that by using a simple ratio the impact exerted by the BMI on the grip strength values obtained will be eliminated/controlled.
My main concern is the strategy used to eliminate (ineffectively) the impact of body size in determining muscle strength levels assessed by handgrip. Further details will be added to the method section..
- "A previous study demonstrated that relative handgrip strength is associated with metabolic diseases [13]. However, there are few studies that examine relative handgrip strength and NAFLD".
Reviewer: I couldn't understand… are there few or no studies that investigated this interrelationship? If there are no other studies described in the literature whose objective was to investigate the association between handgrip strength and NAFLD, please add information regarding the relevance of investigating this topic (Who can benefit? How can such information contribute in terms of public health? What literature gap will the results of this study add?).
- "Therefore, we aimed to examine the association between relative handgrip strength and revalence of NAFLD in adults without HBV/HCV or excessive alcohol uptake through an analysis of a nationwide cross-sectional database over 5 years".
Reviewer: Again, what does HBV/HCV mean? Why are HBV/HCV positive individuals excluded from testing? What about excessive alcohol uptake? Such interrelationship must be contemplated in the introduction section.
MATERIALS AND METHODS
-“"In this cross-sectional study, we analysed the Korean National Health and Nutrition Examination Surveys (KNHANES) data from 2014 to 2018, retrospectively. The KNHANES is performed by the Korea Centres for Disease Control and Prevention in the Ministry of Health and Welfare, which has previously reported with the details [5]”
Reviewer: Please, although the information regarding the macroproject whose data used in this study are derived is available in the literature, additional information regarding the study should be added (Location, period, how the information was collected, objective of the macroproject, sampling).
- “32,704 participants, we excluded subjects who met the following criteria: (1) age ≤ 19 years; (2) absence of handgrip strength data; (3) alcohol consumption per week > 140g for men and > 70g for women [14]; (4) HBV carrier or HCV carrier. After inspecting the exclusion criteria, 19,852 subjects were included in our analysis. All participants signed an informed consent form”
Reviewer: Again, the abbreviations must be preceded by the meaning (HBV Carrier or HCV Carrier). Additionally, what is the reason for excluding subjects who had such characteristics? For this reason I think it is pertinent to add such information in the introduction section.
- "WC was examined based on the midpoint of the lower rib and upper iliac ridge" Reviewer: Add reference for using this specific point.
- "SBP was measured on the right arm using a standard mercury sphygmomanometer".
Reviewer: What are the conditions for carrying out these measurements (number of measurements? Quiet and peaceful environment? Empty bladder? Were those evaluated at rest)?
- “Regular exercise was defined as performing either moderate or vigorous intensity exercise more than three times per week"
Reviewer: Was the questionnaire used validated? How was intensity related to physical activity determined?
- "Hypertension was diagnosed as systolic blood pressure ≥ 140 mmHg, diastolic blood pressure ≥ 90 mmHg, or taking antihypertensive medications"
Reviewer: In the paragraph above, the authors describe that only systolic blood pressure was measured, and here in this paragraph, the authors state that in addition to information regarding the use of medication, they used systolic blood pressure and diastolic blood pressure results to determine hypertension. After all, was the diastolic blood pressure measured or not? Please explain.
- “Diabetes was defined as 8-hour fasting glucose levels ≥ 126 mg/dL, glycated haemoglobin level ≥ 6.5%, taking antidiabetic medications including insulin, or diagnosis by a physician."
Reviewer: Sorry, I don't understand what was done here. Has measured information regarding glycated hemoglobin been obtained? Has information regarding history of illnesses and drug use been collected?
- “Non-alcoholic fatty liver disease (NAFLD) was diagnosed by a well-validated fatty liver predictor, the hepatic steatosis index (HSI), calculated as 8 * alanine aminotransferase (ALT) / aspartate aminotransferase (AST) + BMI (+2 if diabetes; +2 if female). NAFLD was defined as HIS > 36 [18]"
Reviewer: Information regarding the collection of information related to alanine aminotransferase and aspartate aminotransferase was not described in the method. I suggest adding such information.
Reviewer: Authors should pay more attention to the wording of the information used in the study.
- “The weakest relative handgrip strength group (Q1) was considered as the reference group. Multivariate logistic regression analysis was performed to calculate the odds ratios (ORs) and 95% confidence intervals (95% CIs) of NAFLD for relative handgrip strength after adjusting for the confounding factors across relative handgrip strength quartiles.
Reviewer: As far as I know, multinomial means the dependent variable (outcome) has more than 2 levels, multivariate means there is more than one dependent variable (outcome). In this sense, as described by the authors, the analysis used was multinomial logistic regression (in addition to linear regression) l. If yes, please correct!
- "Weight values were applied to all variables."
Reviewer: For this reason, further information regarding the sampling that involved the recruitment/sample of subjects evaluated in the study is necessary to be included in the method section.
-"Handgrip strength was measured three times using a digital grip strength dynamometer (TKK5401; Takei Scientific Instruments Co, Ltd, Tokyo, Japan) [17]. It was measured with the subjects in a standing position and with the arms in a full extension. The participants were instructed to squeeze the dynamometer with as much force as possible, for at least three seconds, three times with each hand alternatively. A rest interval of one minute was given between each trial. Absolute handgrip strength was defined as the summation of the maximum value from each hand and was expressed in kilograms. Relative handgrip strength was calculated as the absolute handgrip strength divided by BMI. The relative handgrip strengths were divided into sex-specific quartiles."
Reviewer: This is my main concern in relation to this study. As described in the literature and by the authors, the assessment of muscle strength through handgrip with a dynamometer has been described as an easy-to-apply measure, with great reliability and whose results can be used both in the sporting and clinical context (Cruz-Jentoft et al., 2019; García-Hermoso et al., 2019; McGrath et al., 2018). However, body size has a determinant impact on muscle strength results measured by handgrip strength (larger and heavier individuals will have higher absolute values of handgrip strength when compared to those with smaller body size and weight) (Nevill et al., 2021). For this reason, different measures have been used in an attempt to eliminate the impact exerted by body size on the absolute values of handgrip strength, including normalization through simple ratio or ratio of proportion (Nevill et al., 2021). However, there is a growing debate regarding the real impact of using these strategies to obtain muscle strength values that are not correlated with body size, and, therefore, more accurate (Külkamp et al., 2020; Nevill et al., 1992; Nevill et al., 2021; Tanner, 1949). This aspect is relevant considering that, in addition to being directly associated with muscle strength and handgrip strength, body size and its body-related indexes are also directly associated with cardiometabolic variables and adverse health conditions, in which, when using inaccurate handgrip strength values, it is not possible to know whether the results obtained are in fact due to the association between the dependent and independent variable, or whether body size is responsible for mediating such association (Diez-Fernandez et al., 2015).
That said, some issues should be considered by the authors of this study:
- For the use of handgrip strength normalized by body-related indexes (e.g. BMI, body weight, body), first of all it is necessary to test whether in fact the normalized handgrip strength variable is in fact not correlated to body size. For example, test a simple correlation or even simple linear regression between absolute values of handgrip strength and variables related to body size, including body weight and height or BMI. Possibly such variables will be correlated (correlation analysis) or statistically associated (linear regression analysis). After this procedure, test whether the normalized variable (handgrip strength / BMI) is related to body size (height, body weight) through correlation or simple linear regression. Certainly what will be identified will be one or all of the variables related to body size will continue to be significant to the handgrip strength index created, only with a change in the direction of these associations.
- I think it is important to demonstrate these results in supplementary documents, to justify the use of more robust techniques in relation to reducing the impact of body size on handgrip strength values in the analyses.
- After identifying that the variables related to body size are still associated with normalized handgrip strength, it will be necessary to use other strategies described in the literature so that you can create handgrip strength indexes not correlated with body size, and thus, more accurate. In this sense, allometry has been described as an adequate method to accommodate body size in determining muscle strength values, and, therefore, should be used. Another option is the use of handgrip strength indexes not correlated with body-related indexes (height, weight, BMI), generated from the use of residuals derived from regression analysis (including absolute values of handgrip strength as outcome and body-related indexes as exposure variable). Logically, the handgrip strength indices created should be tested in association with the body-related index to confirm whether the procedures used eliminated the body size from the obtained handgrip strength values.
Below are some references regarding the content discussed above, and how such procedures may be carried out.
Batterham, A.M., George, K.P.J.J.o.A.P., 1997. Allometric modeling does not determine a dimensionless power function ratio for maximal muscular function. 83:2158-66.
Batterham, A.M., Tolfrey, K., George, K.P., 1997. Nevill's explanation of Kleiber's 0.75 mass exponent: an artifact of collinearity problems in least squares models? Journal of Applied Physiology 82:693-97.
Folland, J.P., Mc Cauley, T.M., Williams, A.G., 2008. Allometric scaling of strength measurements to body size. European journal of applied physiology 102:739-45.
Jaric, S., 2002. Muscle strength testing. Sports Medicine 32:615-31.
Jaric, S., Mirkov, D., Markovic, G.J.T.J.o.S., Research, C., 2005. Normalizing physical performance tests for body size: aproposal for standardization. Journal of Strength and Conditioning Research 19:467-74.
Vanderburgh, P.M., Katch, F.I., Schoenleber, J., Balabinis, C.P., Elliott, R., 1996. Multivariate allometric scaling of men's world indoor rowing championship performance. Medicine Science in Sports Exercise 28:626-30.
Vanderburgh, P.M., Mahar, M.T., Chou, C.H.J.R.q.f.e., sport, 1995. Allometric scaling of grip strength by body mass in college-age men and women. 66:80-84.
Thus, considering that the results will be impacted depending on the strategy used, the results and discussion of findings sections will only be evaluated after following the suggestions made.
REFERENCES
Cruz-Jentoft, A.J., Bahat, G., Bauer, J., Boirie, Y., Bruyère, O., Cederholm, T., Cooper, C., Landi, F., Rolland, Y., et al., 2019. Sarcopenia: revised European consensus on definition and diagnosis. 48:16-31.
Diez-Fernandez, A., Sanchez-Lopez, M., Gulias-Gonzalez, R., Notario-Pacheco, B., Canete Garcia-Prieto, J., Arias-Palencia, N., Martinez-Vizcaino, V.J.P.O., 2015. BMI as a mediator of the relationship between muscular fitness and cardiometabolic risk in children: a mediation analysis. 10:e0116506.
García-Hermoso, A., Ramírez-Campillo, R., Izquierdo, M.J.S.M., 2019. Is muscular fitness associated with future health benefits in children and adolescents? A systematic review and meta-analysis of longitudinal studies. 49:1079-94.
Külkamp, W., Ache-Dias, J., Kons, R.L., Detanico, D., Dal Pupo, J., 2020. The ratio standard is not adequate for scaling handgrip strength in judo athletes and nonathletes. Journal of Exercise Rehabilitation 16:175.
McGrath, R.P., Kraemer, W.J., Al Snih, S., Peterson, M.D.J.S.m., 2018. Handgrip strength and health in aging adults. 48:1993-2000.
Nevill, A.M., Ramsbottom, R., Williams, C.J.E.j.o.a.p., physiology, o., 1992. Scaling physiological measurements for individuals of different body size. 65:110-17.
Nevill, A.M., Tomkinson, G.R., Lang, J.J., Wutz, W., Myers, T.D.J.M., sports, s.i., exercise, 2021. How Should Adult Handgrip Strength Be Normalized? Allometry Reveals New Insights and Associated Reference Curves.
Tanner, J., 1949. Fallacy of per-weight and per-surface area standards, and their relation to spurious correlation. Journal of applied physiology 2:1-15.
Author Response
Response to Reviewer 1 comments
We greatly appreciate your thorough review of manuscript. We have revised the paper based on the reviewer`s suggestions. We hope that our revisions meet your requirements and that the manuscript is deemed suitable for publication in the IJERPH
ABSTRACT
1. Please add briefly the information of the investigated sample (age group, gender, place where the data were collected).
Answers) Thank you for your valuable comments. As your comments, we should have described the information of the participants in details. We have added the information of the organization where the data were collected in the abstract section, and the detailed information of the participants in the abstract and result section.
|
Line 16, Abstract section) We investigated the association by using relative handgrip strength in a nationwide cross-sectional survey. The participants were recruited from the Korean National Health and Nutrition Examination Surveys (KNHANES). |
|
Line 22, Abstract section) Multinomial logistic regression was analysed to investigate the association of relative handgrip strength with prevalence of NAFLD. The mean age of study population was 45.8 ± 0.3 in men, and 48.3 ± 0.2 in women. The proportion of males was 37.5%. |
|
Line 144, Result section) Table 1 shows the baseline characteristics of the 19,852 participants (7,443 men, and 12,409 women) according to relative handgrip strength quartile. The mean age of the men and women was 45.8 ± 0.3 and 48.3 ± 0.2 years respectively, with 37.5% of the sample being male. |
2. I suggest describing in this section how the handgrip strength was normalized and what body-related index was used for such a procedure.
Answers) Thank you for your thorough review. We agree with your comment that detailed explanation of process of normalizing handgrip strength and body-related index in abstract section. Absolute handgrip strength was divided by BMI, which is called relative handgrip strength. The index was widely used to avoid the potential bias effect of BMI on the estimation of handgrip strength (Ji C et al, 2020).
|
Line 19, Abstract section) We used normalized handgrip strength, which is called relative handgrip strength. The index was defined as handgrip strength divided by BMI. These subjects were divided into quartile groups according to relative handgrip strength. |
|
Reference for reviewer) 1. Ji C, Xia Y, Tong S, Wu Q, Zhao Y. Association of handgrip strength with the prevalence of metabolic syndrome in US adults: the national health and nutrition examination survey. Aging (Albany NY). 2020;12(9):7818–29. Epub 2020/05/05. |
3. Please describe the words in full before adding abbreviations.
Ex: For those who are reading the work for the first time, the acronyms NAFLD or HBV/HCV will have no meaning. Please add such information!
Answers) We appreciate your sharp comment. We didn`t describe the in full name before using abbreviations. As you commented, we have used the term, NAFLD, in full when used for the first time. On the other hand, we have deleted HBV/HCV, and replaced them with hepatitis B surface antigen or hepatitis C antibody.
|
Line 14, Abstract section) Sarcopenia is known to be associated with non-alcoholic fatty liver disease (NAFLD). However, few studies have revealed the association between muscle strength and prevalence of NAFLD. |
|
Line 72, Methods section / Study population subsection) (4) positive for hepatitis B surface antigen or hepatitis C antibody [21] |
4. - “As handgrip strength increased, the age, waist circumference (WC), fasting glucose, systolic BP, hypertension, diabetes, and hepatic steatosis index (HSI) are significantly decreased”.
I was unable to verify the relevance of including the aforementioned sentence in the “abstract” section of this study. In my opinion, such sentence can be deleted from this section.
Answers) As you commented, we have deleted the aforementioned sentence in the abstract section.
5. As well as logistic regression values, I suggest adding association values referring to multiple linear regression analysis (HSI index).
Answers) As your recommendation, we specified the values of multiple linear regression analysis, and added the values in the abstract section.
|
Line 24, Abstract section) In multiple linear regression, relative handgrip strength was negatively associated with HSI index (Standardized β = -0.70; standard error (SE), 0.08; P < 0.001 in men, Standardized β = -0.94; standard error (SE), 0.07; P < 0.001 in women). |
6. - "Relative handgrip strength, used as a biomarker of sarcopenia, is independently negatively associated with NAFLD".
Given the nature of the investigated associations (cross-sectional design), I think it is not coherent to add vocabulary that denotes causality (e.g. negatively associated). I suggest modifying the term used (e.g., inversely associated).
Answers) Thank you for your sharp comment. We can define the negative association only if the causality of handgrip strength and NAFLD has been identified. However, it can`t be identified because the data were cross-sectional. Consequently, we corrected “negatively associated” to “inversely associated”.
7. - "The relationship between handgrip strength and NAFLD was independent of metabolic syndrome".
In my opinion, since the relevance of the interrelationship of NAFLD x metabolic syndrome was never described, adding in the concluding section that the results were identified independently of metabolic syndrome does not seem to make sense to me. if the authors choose to maintain such a sentence, they must substantiate why this finding is relevant (results independent of metabolic syndrome). Otherwise, I suggest deleting!
Answers) NAFLD and metabolic syndrome can be said to be “independent” if incidence of NAFLD remains the same regardless of prevalence of metabolic syndrome. In other words, metabolic syndrome does not affect incidence of NAFLD. Therefore, we delete the aforementioned sentence because the correlation of NAFLD and metabolic syndrome were not analyzed.
8. - “Furthermore, its association may be mechanically linked through insulin resistance and inflammation”.
Do not include justification for findings in the concluding section. I suggest deleting!
Answers) As your comment, we delete the sentence because it was not suitable in the context. The inference of the relationship mediated by insulin resistance and inflammation was described in discuss section to explain mechanism of association between handgrip strength and NAFLD
INTRODUCTION
9. - “Sarcopenia, a condition of decreasing skeletal muscle mass and strength, is a major health concern worldwide”.
Please reference such information.
Answers) For supporting the information for definition and significance of sarcopenia, we have added the reference.
|
Line 286, References section) 1. Beaudart, C.; McCloskey, E.; Bruyère, O.; Cesari, M.; Rolland, Y.; Rizzoli, R.; de Carvalho, I.A.; Thiyagarajan, J.A.; Bautmans, I.; Bertière, M.-C. Sarcopenia in daily practice: assessment and management. BMC geriatrics 2016, 16, 1-10. |
10. - “Nevertheless, few studies on the association between muscle strength and NAFLD have been reported”.
Could you please cite two or three studies that support this interrelationship?
Answers) There are a few articles for the association between muscle strength and NAFLD. The previous studies suggested that muscle strength was inversely associated with NAFLD and the mechanism of the association was linked through insulin resistance and inflammation (Ce Fre CH, et al., 2019; Hao, Wang et al. 2020). However, these studies recruited a small number of subjects. Our study reinforced the relationship collecting a large number of participants over 5 years. As your recommendation, we have cited the references in the sentence.
|
Line 308, References section) 9. De Fre CH, De Fre MA, Kwanten WJ et al: Sarcopenia in patients with nonalcoholic fatty liver disease: Is it a clinically significant entity? Obes Rev, 2019; 20(2): 353–63 10. Hao, L., Z. Wang, Y. Wang, J. Wang and Z. Zeng (2020). "Association between cardiorespiratory fitness, relative grip strength with non-alcoholic fatty liver disease." Medical science monitor: international medical journal of experimental and clinical research 26: e923015-923011. |
11. - “With the emerging importance of handgrip strength, recent studies have also used relative handgrip strength (absolute handgrip strength/BMI), which is an independent indicator for BMI.”
Please add references that by using a simple ratio the impact exerted by the BMI on the grip strength values obtained will be eliminated/controlled.
My main concern is the strategy used to eliminate (ineffectively) the impact of body size in determining muscle strength levels assessed by handgrip. Further details will be added to the method section.
Answers) Previous studies suggest that handgrip strength normalized for BMI can be an effective tool to quantifying muscle quality. We have added the references in the sentence. However, although relative handgrip strength is the index adjusted by BMI, there are no studies that the index is independent factor for BMI. Therefore, we have analysed the association between handgrip strength and body size by using linear regression and described the results in the last comment.
|
Line 328, References section) 16. Choquette, S.; Bouchard, D.; Doyon, C.; Sénéchal, M.; Brochu, M.; Dionne, I.J. Relative strength as a determinant of mobility in elders 67–84 years of age. a nuage study: nutrition as a determinant of successful aging. The journal of nutrition, health & aging 2010, 14, 190-195. 17. Straight, C.; Brady, A.; Schmidt, M.; Evans, E. Comparison of laboratory-and field-based estimates of muscle quality for predicting physical function in older women. J Aging Res Clin Pract 2013, 2, 276-279. |
12. - "A previous study demonstrated that relative handgrip strength is associated with metabolic diseases [13]. However, there are few studies that examine relative handgrip strength and NAFLD".
I couldn't understand… are there few or no studies that investigated this interrelationship? If there are no other studies described in the literature whose objective was to investigate the association between handgrip strength and NAFLD, please add information regarding the relevance of investigating this topic (Who can benefit? How can such information contribute in terms of public health? What literature gap will the results of this study add?).
Answers) Thank you for your considerate comment. Although there are few studies that examine relative handgrip strength and NAFLD, we didn`t cite references in the literature. As your recommendation, we have cited the references in the sentence.
|
Line 310, References section) 10. Hao, L., Z. Wang, Y. Wang, J. Wang and Z. Zeng (2020). "Association between cardiorespiratory fitness, relative grip strength with non-alcoholic fatty liver disease." Medical science monitor: international medical journal of experimental and clinical research 26: e923015-923011. 17. Xia, Y.; Cao, L.; Liu, Y.; Wang, X.; Zhang, S.; Meng, G.; Zhang, Q.; Liu, L.; Wu, H.; Gu, Y. Longitudinal Associations Between Hand Grip Strength and Non-Alcoholic Fatty Liver Disease in Adults: A Prospective Cohort Study. Frontiers in Medicine 2021, 8 |
13. - "Therefore, we aimed to examine the association between relative handgrip strength and revalence of NAFLD in adults without HBV/HCV or excessive alcohol uptake through an analysis of a nationwide cross-sectional database over 5 years".
Again, what does HBV/HCV mean? Why are HBV/HCV positive individuals excluded from testing? What about excessive alcohol uptake? Such interrelationship must be contemplated in the introduction section.
Answers) Thank you for your sharp comment. “without HBV/HCV” is too vague to understand. In KNHANES data, there are serological marker for hepatitis B and hepatitis C. When the serological markers for hepatitis B or hepatitis C were positive, the subjects were excluded in our study. The reason of excluding the participants whose markers were positive is that NAFLD is defined after the exclusion of other potential causes such as alcohol, viral infections, and drugs (Campos-Murguía A et al.). For this reason, the participants who drank excessive alcohol were also excluded in our study. Consequently, we deleted the words “without HBV/HCV or excessive alcohol uptake” in the introduction section because it is out of context. Instead, we have changed “without HBV/HCV” to “(4) positive for hepatitis B surface antigen or hepatitis C antibody.” in methods section and added the reference for the definition of NAFLD.
|
Line 69, Methods section / Study population subsection) Among a total of the 32,704 participants, we excluded subjects who met the following criteria: (1) age ≤ 19 years; (2) absence of handgrip strength data; (3) alcohol consumption per week > 140g for men and > 70g for women [20]; (4) positive for hepatitis B surface antigen or hepatitis C antibody [21]. After inspecting the exclusion criteria, 19,852 subjects were included in our analysis. All participants signed an informed consent form. |
|
Line 342, References section) 21. Campos-Murguía A, Ruiz-Margáin A, González-Regueiro JA, et al. Clinical assessment and management of liver fibrosis in non-alcoholic fatty liver disease. World J Gastroenterol 2020; 26: 5919–5943 |
MATERIALS AND METHODS
14. - “In this cross-sectional study, we analysed the Korean National Health and Nutrition Examination Surveys (KNHANES) data from 2014 to 2018, retrospectively. The KNHANES is performed by the Korea Centres for Disease Control and Prevention in the Ministry of Health and Welfare, which has previously reported with the details [5]”
Please, although the information regarding the macroproject whose data used in this study are derived is available in the literature, additional information regarding the study should be added (Location, period, how the information was collected, objective of the macroproject, sampling).
Answers) The KNHANES is a national supervision system that has been evaluating the health and nutritional status of Koreans since 1998. It is a continuous survey with nationally representative samples of Korea. Based upon the National Health Promotion Act, the surveys have been handled by the Korea Centers for Disease Control and Prevention in the Ministry of Health and Welfare. This cross-sectional survey contains approximately 10,000 participants each year as a sample and collects information on socioeconomic status, health-associated behaviours, quality of life, healthcare utilization, anthropometric measures, biochemical and clinical profiles for non-communicable diseases with three component surveys: health interview, and health examination. The health interview and examination are conducted by trained staff members, including physicians, medical technicians and health interviewers, at a mobile examination centre (Kweon S et al., 2014).
|
Line 65, Methods section / Study population subsection) The KNHANES is a national surveillance system to assess the health and nutritional status of Koreans and performed by the Korea Centres for Disease Control and Prevention in the Ministry of Health and Welfare, which has previously reported with the details. |
|
Reference for reviewer) 1. Kweon S, et al. Data resource profile: The Korea National Health and Nutrition Examination Survey (KNHANES) Int. J. Epidemiol. 2014;43:69–77. doi: 10.1093/ije/dyt228 |
15. - “32,704 participants, we excluded subjects who met the following criteria: (1) age ≤ 19 years; (2) absence of handgrip strength data; (3) alcohol consumption per week > 140g for men and > 70g for women [14]; (4) HBV carrier or HCV carrier. After inspecting the exclusion criteria, 19,852 subjects were included in our analysis. All participants signed an informed consent form”
Again, the abbreviations must be preceded by the meaning (HBV Carrier or HCV Carrier). Additionally, what is the reason for excluding subjects who had such characteristics? For this reason I think it is pertinent to add such information in the introduction section.
Answers) As you commented, the abbreviations have been described in full terms before using it. Moreover, we have changed the words “HBV carrier or HCV carrier” (4) positive for hepatitis B surface antigen or hepatitis C antibody. Additionally, we excluded the subjects whose serologic markers were positive because NAFLD is defined as the accumulation of fat in the liver (>5%), after the exclusion of other potential causes such as alcohol, viral infections, and drugs (Campos-Murguía A et al.).
|
Line 72, Methods section / Study population subsection) (4) positive for hepatitis B surface antigen or hepatitis C antibody [21]. |
|
Line 342, References section) 21. Campos-Murguía A, Ruiz-Margáin A, González-Regueiro JA, et al. Clinical assessment and management of liver fibrosis in non-alcoholic fatty liver disease. World J Gastroenterol 2020; 26: 5919–5943 |
16. - "WC was examined based on the midpoint of the lower rib and upper iliac ridge"
Add reference for using this specific point.
Answers) Previous studies used the same point to measure WC. Previous study measured the WC using a flexible tape (Seca 220; Seca), and determined at the midpoint the lowest margin of the rib and the uppermost border of the iliac crest during expiration(Lee and Song 2020)(Lee and Song 2020). The other study also examined the WC in the same way; the midpoint between the lowest rib and upper iliac spine(Kim and Park 2020)(Kim and Park 2020). In order to support the way to measure, we add the reference in references section.
|
Line 78, Methods section / Anthropometric and laboratory measurements subsection) WC was measured by using a flexible tape (Seca 220; Seca), and examined based on the midpoint the lowest margin of the rib and the uppermost border of the iliac crest during expiration [22,23]. |
|
Line 345, References section) 22. Lee, J.S.; Song, Y.H. Relationship between waist circumference and cardiovascular risk factors in adolescents: analysis of the Korea National Health and Nutrition Examination Survey Data. Korean Circulation Journal 2020, 50, 723-732. 23. Kim, B.; Park, E.Y. The combined effect of socioeconomic status and metabolic syndrome on depression: the Korean National Health and Nutrition Examination Survey (KNHANES). BMC public health 2020, 20, 1-12. |
17. - "SBP was measured on the right arm using a standard mercury sphygmomanometer".
What are the conditions for carrying out these measurements (number of measurements? Quiet and peaceful environment? Empty bladder? Were those evaluated at rest)?
Answers) In the KNHANES, BP (SBP and DBP) was examined using a mercury sphygmomanometer after the participants had rested for 5 minutes in a sitting position [Baumanometer Wall Unit 33(0850)]. All BP examinations were conducted on the right arm three times using the same instruments at 30-second intervals with a cuff appropriate for arm circumference. As your recommendation, we have added the measurement of blood pressure in the methods section.
|
Line 80, Methods section / Anthropometric and laboratory measurements subsection) All BP were measured using a mercury sphygmomanometer after the participants had rested for 5 minutes in a sitting position [Baumanometer Wall Unit 33(0850)]. All BP examinations were conducted on the right arm three times using the same instruments at 30-second intervals with a cuff appropriate for arm circumference [24]. |
|
Line 350, References section) 24. Kang, M.G.; Kim, K.H.; Koh, J.S.; Park, J.R.; Hwang, S.J.; Hwang, J.Y.; Ahn, J.H.; Jang, J.Y.; Jeong, Y.H.; Kwak, C.H. Association between pulse pressure and body mass index in hypertensive and normotensive populations in the Korea National Health and Nutrition Examination Survey V, 2010–2012. The Journal of Clinical Hypertension 2017, 19, 395-401. |
18. - “Regular exercise was defined as performing either moderate or vigorous intensity exercise more than three times per week"
Was the questionnaire used validated? How was intensity related to physical activity determined?
Answers) The Global Physical Activity Questionnaire (GPAQ) was standardized to measure the level of physical activity of people throughout the world. The GPAQ consisted of a total 16 items regarding the amount of time spent on vigorous physical activity, and moderate physical activity (#33872483). The intensity of the physical activity was measured by the time of physical activity performed per week (number of days per week). The questions on vigorous and moderate physical activity in KNHANES were as follows:
1) Questions on vigorous physical activity: On how many days in the past week did you perform vigorous physical activity that made you feel very short of breath, or heart beating much faster than usual for at least 10 min?
Examples of vigorous physical activities: running, mountain climbing, fast cycling, fast swimming, soccer, basketball and carrying heavy objects,
2) Questions on moderate physical activity: On how many days in the past week did you perform moderate physical activity that made you feel a little short of breath, or heart beating a little faster than usual for at least 10 min?
Examples of moderate physical activities: slow swimming, babysitting, cleaning, moving and carrying light objects
We have added the information in the methods section and references section.
|
Line 91, Methods section / Anthropometric and laboratory measurements subsection) The Global Physical Activity Questionnaire (GPAQ) was standardized to measure the level of physical activity of people [25]. |
|
Line 354, References section: 25. Kim, S.; Choi, S.; Kim, J.; Park, S.; Kim, Y.; Park, O.; Oh, K. Trends in health behaviors over 20 years: findings from the 1998-2018 Korea National Health and Nutrition Examination Survey. Epidemiology and health 2021, 43. |
19. - "Hypertension was diagnosed as systolic blood pressure ≥ 140 mmHg, diastolic blood pressure ≥ 90 mmHg, or taking antihypertensive medications"
In the paragraph above, the authors describe that only systolic blood pressure was measured, and here in this paragraph, the authors state that in addition to information regarding the use of medication, they used systolic blood pressure and diastolic blood pressure results to determine hypertension. After all, was the diastolic blood pressure measured or not? Please explain.
Answers) Thank you for your sharp comment. In the KNHANES, both systolic blood pressure and diastolic blood pressure were measured. Furthermore, taking antihypertensive medications could be checked based on questionnaires (Kang, M.G. et al., 2017). For reflecting your comment, we describe diastolic blood pressure and systolic blood pressure.
|
Line 77, Methods section / Anthropometric and laboratory measurements subsection) Data included age, waist circumference (WC), and systolic blood pressure (SBP) and diastolic blood pressure (DBP). |
|
Reference for reviewer) 1. Kang, M.G.; Kim, K.H.; Koh, J.S.; Park, J.R.; Hwang, S.J.; Hwang, J.Y.; Ahn, J.H.; Jang, J.Y.; Jeong, Y.H.; Kwak, C.H. Association between pulse pressure and body mass index in hypertensive and normotensive populations in the Korea National Health and Nutrition Examination Survey V, 2010–2012. The Journal of Clinical Hypertension 2017, 19, 395-401. |
20. - “Diabetes was defined as 8-hour fasting glucose levels ≥ 126 mg/dL, glycated haemoglobin level ≥ 6.5%, taking antidiabetic medications including insulin, or diagnosis by a physician."
Sorry, I don't understand what was done here. Has measured information regarding glycated hemoglobin been obtained? Has information regarding history of illnesses and drug use been collected?
Answers) Glycated haemoglobin (HbA1c) has been collected in the KNHANES. HbA1c levels were decided by performance liquid chromatography using an automated HLC-723G7 analyzer (Tosoh Corporation, Tokyo, Japan). Taking antidiabetic medications and diagnosis by a physician have been obtained based on questionnaires in the data. We have added this explanation in the methods section.
|
Line 95, Methods section / Anthropometric and laboratory measurements subsection) Diabetes was defined as 8-hour fasting glucose levels ≥ 126 mg/dL, glycated haemoglobin (HbA1c) level ≥ 6.5%, taking antidiabetic medications including insulin, or diagnosis by a physician [27]. HbA1c levels were decided by performance liquid chromatography using an automated HGLC-723G7 analyzer (Tosoh Corporation, Tokyo, Japan) [28]. The medication history and diagnosis by physician have been obtained based on questionnaires in the data. |
|
Line 362, References section) 28. Oh, I.H.; Park, J.H.; Lee, C.H.; Park, J.-S. The association of normal range glycated hemoglobin with restrictive lung pattern in the general population. PLoS One 2015, 10, e0117725. |
21. - “Non-alcoholic fatty liver disease (NAFLD) was diagnosed by a well-validated fatty liver predictor, the hepatic steatosis index (HSI), calculated as 8 * alanine aminotransferase (ALT) / aspartate aminotransferase (AST) + BMI (+2 if diabetes; +2 if female). NAFLD was defined as HIS > 36 [18]"
Information regarding the collection of information related to alanine aminotransferase and aspartate aminotransferase was not described in the method. I suggest adding such information.
Answers) Thank you for your sharp comment. ALT and AST were sampled from venous blood and measured using Hitachi Automatic analyzer 7600-210 (Hitachi, Tokyo, Japan), which was the same method as fasting plasma blood glucose, TC, and TG). We have added the description in the methods section.
|
Line 100, Methods section / Anthropometric and laboratory measurements subsection) Fasting plasma blood glucose, total cholesterol (TC), and triglyceride (TG), alanine aminotransferase (ALT), and aspartate aminotransferase (AST) were measured using the Hitachi Automatic Analyzer 7600-210 (Hitachi, Tokyo, Japan) [28]. |
|
Line 362, References section) 28. Oh, I.H.; Park, J.H.; Lee, C.H.; Park, J.-S. The association of normal range glycated hemoglobin with restrictive lung pattern in the general population. PLoS One 2015, 10, e0117725. |
22. Authors should pay more attention to the wording of the information used in the study.
Answers) We really apologize for our erratum. We have changed HIS to HSI.
23. - “The weakest relative handgrip strength group (Q1) was considered as the reference group. Multivariate logistic regression analysis was performed to calculate the odds ratios (ORs) and 95% confidence intervals (95% CIs) of NAFLD for relative handgrip strength after adjusting for the confounding factors across relative handgrip strength quartiles.
As far as I know, multinomial means the dependent variable (outcome) has more than 2 levels, multivariate means there is more than one dependent variable (outcome). In this sense, as described by the authors, the analysis used was multinomial logistic regression (in addition to linear regression) l. If yes, please correct!
Answers) Thank you for your sharp comment. As you commented, we categorized relative handgrip strength (outcome) as quartiles. It means we used multinomial logistic regression, not multivariate logistic regression. Therefore, “multivariate” have been amended to “multinomial”.
|
Line 133, Methods section / Statistical analysis subsection) Multinomial logistic regression analysis was performed to calculate the odds ratios (ORs) and 95% confidence intervals (95% CIs) of NAFLD for relative handgrip strength after adjusting for the confounding factors across relative handgrip strength quartiles. |
24. - "Weight values were applied to all variables."
For this reason, further information regarding the sampling that involved the recruitment/sample of subjects evaluated in the study is necessary to be included in the method section.
Answers) Each data of the participants has specific sampling weights value. The values were made for sample participants to represent the Korean population by accounting for the survey non-response, complex survey design and post-stratification. The weights based on the inverse of selection probabilities and inverse of response rates were corrected by adjusting them to the sex- and age-specific Korean populations.
|
Line 136, Methods section / Statistical analysis subsection) Weight values were applied to all variables. The values were made for sample participants to represent the Korean population by accounting for the non-response, complex survey design and post-stratification [34]. |
|
Line 376, References section) 34. Kweon, S.; Kim, Y.; Jang, M.-j.; Kim, Y.; Kim, K.; Choi, S.; Chun, C.; Khang, Y.-H.; Oh, K. Data resource profile: The Korea national health and nutrition examination survey (KNHANES). International journal of epidemiology 2014, 43, 69-77. |
25. - "Handgrip strength was measured three times using a digital grip strength dynamometer (TKK5401; Takei Scientific Instruments Co, Ltd, Tokyo, Japan) [17]. It was measured with the subjects in a standing position and with the arms in a full extension. The participants were instructed to squeeze the dynamometer with as much force as possible, for at least three seconds, three times with each hand alternatively. A rest interval of one minute was given between each trial. Absolute handgrip strength was defined as the summation of the maximum value from each hand and was expressed in kilograms. Relative handgrip strength was calculated as the absolute handgrip strength divided by BMI. The relative handgrip strengths were divided into sex-specific quartiles."
This is my main concern in relation to this study. As described in the literature and by the authors, the assessment of muscle strength through handgrip with a dynamometer has been described as an easy-to-apply measure, with great reliability and whose results can be used both in the sporting and clinical context (Cruz-Jentoft et al., 2019; García-Hermoso et al., 2019; McGrath et al., 2018). However, body size has a determinant impact on muscle strength results measured by handgrip strength (larger and heavier individuals will have higher absolute values of handgrip strength when compared to those with smaller body size and weight) (Nevill et al., 2021). For this reason, different measures have been used in an attempt to eliminate the impact exerted by body size on the absolute values of handgrip strength, including normalization through simple ratio or ratio of proportion (Nevill et al., 2021). However, there is a growing debate regarding the real impact of using these strategies to obtain muscle strength values that are not correlated with body size, and, therefore, more accurate (Külkamp et al., 2020; Nevill et al., 1992; Nevill et al., 2021; Tanner, 1949). This aspect is relevant considering that, in addition to being directly associated with muscle strength and handgrip strength, body size and its body-related indexes are also directly associated with cardiometabolic variables and adverse health conditions, in which, when using inaccurate handgrip strength values, it is not possible to know whether the results obtained are in fact due to the association between the dependent and independent variable, or whether body size is responsible for mediating such association (Diez-Fernandez et al., 2015).
That said, some issues should be considered by the authors of this study:
- For the use of handgrip strength normalized by body-related indexes (e.g. BMI, body weight, body), first of all it is necessary to test whether in fact the normalized handgrip strength variable is in fact not correlated to body size. For example, test a simple correlation or even simple linear regression between absolute values of handgrip strength and variables related to body size, including body weight and height or BMI. Possibly such variables will be correlated (correlation analysis) or statistically associated (linear regression analysis). After this procedure, test whether the normalized variable (handgrip strength / BMI) is related to body size (height, body weight) through correlation or simple linear regression. Certainly what will be identified will be one or all of the variables related to body size will continue to be significant to the handgrip strength index created, only with a change in the direction of these associations.
- I think it is important to demonstrate these results in supplementary documents, to justify the use of more robust techniques in relation to reducing the impact of body size on handgrip strength values in the analyses.
- After identifying that the variables related to body size are still associated with normalized handgrip strength, it will be necessary to use other strategies described in the literature so that you can create handgrip strength indexes not correlated with body size, and thus, more accurate. In this sense, allometry has been described as an adequate method to accommodate body size in determining muscle strength values, and, therefore, should be used. Another option is the use of handgrip strength indexes not correlated with body-related indexes (height, weight, BMI), generated from the use of residuals derived from regression analysis (including absolute values of handgrip strength as outcome and body-related indexes as exposure variable). Logically, the handgrip strength indices created should be tested in association with the body-related index to confirm whether the procedures used eliminated the body size from the obtained handgrip strength values.
Answers) Thank you for your valuable comment. In order to find more reliable index, we have analysed the association between handgrip strength and body size in various ways. The previous study you commented suggested handgrip strength divided height2 is also an effective tool to normalize strength (Nevill et al., 2021) as well as relative handgrip strength (absolute handgrip strength/BMI). Therefore, we evaluated handgrip strength in the three ways: (1) absolute handgrip strength (AGS); (2) relative handgrip strength (AGS/BMI); (3) normalized handgrip strength (AGS/Height2). Linear regression analysis was conducted to identify the independence of each index of handgrip strength and body size (BMI, weight, height). Supplementary table 1-3 show the association between AGS and body size (BMI, weight, height). AGS was significantly associated with BMI, weight, and height in men and women. Supplementary table 4-6 suggest the association between relative handgrip strength (RGS) and body size (BMI, weight, height). RGS was inversely associated with BMI, and weight in men and women. On the other hands, RGS was significantly to proportional height in men and women. Supplementary 7-9 suggest the relationship of normalized handgrip strength (NGS) and body size (BMI, weight, height). NGS was significantly associated with BMI, weight, and height in men and women. Because HSI, the definition of NAFLD in our study, includes BMI [HSI = 8 * ALT / AST + BMI (+2 if diabetes; +2 if female)], we also use another serologic marker of NAFLD, which is lipid accumulation product (LAP), in order to exclude the impact of BMI on the diagnosis of NAFLD. LAP were calculated as [Waist circumference (cm) – 65] * TG (mmol/L) in men, and [Waist circumference (cm) – 58] * TG (mmol/L) in women. Supplementary Table 10 suggests the relationship of relative handgrip strength and serum biomarker for diagnosing fatty liver (HSI, and LAP). Through the results, we find RGS are inversely associated with both HSI, LAP in men and women. Nevertheless, there is a limitation for the reliability of RGS because it is not independent body size. Therefore, we have added the limitation in the discussion section. Further studies of body-related index are needed in the future.
Supplementary Table 1. Results of linear regression analysis to assess the relationships between absolute handgrip strength and BMI in Koreans
|
|
Men |
|
|
|
|
Standardized β |
SE |
p-value |
|
Unadjusted |
0.096 |
0.007 |
<0.001 |
|
|
|
|
|
|
|
|
Women |
|
|
|
Standardized β |
SE |
p-value |
|
Unadjusted |
0.043 |
0.009 |
<0.001 |
Supplementary Table 2. Results of linear regression analysis to assess the relationships between absolute handgrip strength and body weight in Koreans
|
|
Men |
|
|
|
|
Standardized β |
SE |
p-value |
|
Unadjusted |
0.589 |
0.021 |
<0.001 |
|
|
|
|
|
|
|
|
Women |
|
|
|
Standardized β |
SE |
p-value |
|
Unadjusted |
0.525 |
0.023 |
<0.001 |
Supplementary Table 3. Results of linear regression analysis to assess the relationships between absolute handgrip strength and height in Koreans
|
|
Men |
|
|
|
|
Standardized β |
SE |
p-value |
|
Unadjusted |
0.370 |
0.011 |
<0.001 |
|
|
|
|
|
|
|
|
Women |
|
|
|
Standardized β |
SE |
p-value |
|
Unadjusted |
0.557 |
0.014 |
<0.001 |
Supplementary Table 4. Results of linear regression analysis to assess the relationships between RGS (AGS/BMI) and BMI in Koreans
|
|
Men |
|
|
|
|
Standardized β |
SE |
p-value |
|
Unadjusted |
-2.00 |
0.07 |
<0.001 |
|
|
|
|
|
|
|
|
Women |
|
|
|
Standardized β |
SE |
p-value |
|
Unadjusted |
-3.50 |
0.07 |
<0.001 |
Supplementary Table 5. Results of linear regression analysis to assess the relationships between RGS (AGS/BMI) and body weight in Koreans
|
|
Men |
|
|
|
|
Standardized β |
SE |
p-value |
|
Unadjusted |
-3.11 |
0.26 |
<0.001 |
|
|
|
|
|
|
|
|
Women |
|
|
|
Standardized β |
SE |
p-value |
|
Unadjusted |
-4.57 |
0.20 |
<0.001 |
Supplementary Table 6. Results of linear regression analysis to assess the relationships between RGS (AGS/BMI) and height in Koreans
|
|
Men |
|
|
|
|
Standardized β |
SE |
p-value |
|
Unadjusted |
3.52 |
0.12 |
<0.001 |
|
|
|
|
|
|
|
|
Women |
|
|
|
Standardized β |
SE |
p-value |
|
Unadjusted |
5.72 |
0.12 |
<0.001 |
Supplementary Table 7. Results of linear regression analysis to assess the relationships between RGS (AGS/Height2) and BMI in Koreans
|
|
Men |
|
|
|
|
Standardized β |
SE |
p-value |
|
Unadjusted |
0.318 |
0.021 |
<0.001 |
|
|
|
|
|
|
|
|
Women |
|
|
|
Standardized β |
SE |
p-value |
|
Unadjusted |
0.275 |
0.023 |
<0.001 |
Supplementary Table 8. Results of linear regression analysis to assess the relationships between RGS (AGS/Height2) and body weight in Koreans
|
|
Men |
|
|
|
|
Standardized β |
SE |
p-value |
|
Unadjusted |
1.032 |
0.073 |
<0.001 |
|
|
|
|
|
|
|
|
Women |
|
|
|
Standardized β |
SE |
p-value |
|
Unadjusted |
0.858 |
0.061 |
<0.001 |
Supplementary Table 9. Results of linear regression analysis to assess the relationships between RGS (AGS/Height2) and height in Koreans
|
|
Men |
|
|
|
|
Standardized β |
SE |
p-value |
|
Unadjusted |
0.124 |
0.041 |
0.003 |
|
|
|
|
|
|
|
|
Women |
|
|
|
Standardized β |
SE |
p-value |
|
Unadjusted |
0.201 |
0.042 |
<0.001 |
Supplementary Table 10. Results of analysis of variance to assess relationship between RGS (Quartile) and fatty liver indexes (HSI, LAP) in Koreans
|
Men |
Q1 |
Q2 |
Q3 |
Q4 |
p value |
|
|
|
|
≤ 2.80 |
2.81 – 3.30 |
3.31 – 3.70 |
> 3.70 |
|
|
HSI |
|
36.0 ± 0.2 |
34.3 ± 0.2 |
32.6 ± 0.2 |
30.5 ± 0.1 |
< 0.001 |
|
LAP |
|
48.1 ± 1.2 |
42.1 ± 1.1 |
34.4 ± 1.1 |
24.4 ± 0.7 |
< 0.001 |
|
Women |
Q1 |
Q2 |
Q3 |
Q4 |
p value |
|
|
|
≤ 1.70 |
1.71 – 2.00 |
2.01 – 2.40 |
> 2.40 |
|
|
|
HSI |
|
35.1 ± 0.1 |
33.2 ± 0.1 |
31.2 ± 0.1 |
28.9 ± 0.1 |
< 0.001 |
|
LAP |
|
43.7 ± 0.7 |
31.5 ± 0.7 |
23.3 ± 0.5 |
15.3 ± 0.4 |
< 0.001 |
|
Line 112, Methods section / Assessment of handgrip strength subsection) Handgrip strength is known to be correlated with body size (BMI, weight, height). There are several studies to reduce the effects of body size on handgrip strength, such as hand grip force divided by weight, or height2, or BMI [30,31]. Among these indexes, we used relative handgrip strength that is calculated as the absolute handgrip strength divided by BMI because it is previously used as an indicator for muscle strength [32]. |
|
Line 255, Discussion section) Finally, an appropriate indicator to completely eliminate the effect of body size on handgrip strength has not been determined. Although relative handgrip strength was used to reduce the effect of body size, even relative handgrip strength cannot completely eliminate the effect of body size [30]. In the future, it seems that additional research is needed on muscle strength-related indicators independent of body size, especially in Korean population. |
|
Line 366, References section) 30. Nevill, A.M.; Tomkinson, G.R.; Lang, J.J.; Wutz, W.; Myers, T.D. How Should Adult Handgrip Strength Be Normalized? Allometry Reveals New Insights and Associated Reference Curves. Medicine and science in sports and exercise 2021. 31. Ramírez‐Vélez, R.; Correa‐Bautista, J.E.; García‐Hermoso, A.; Cano, C.A.; Izquierdo, M. Reference values for handgrip strength and their association with intrinsic capacity domains among older adults. Journal of cachexia, sarcopenia and muscle 2019, 10, 278-286. 32. Cho, J.; Lee, I.; Park, D.-H.; Kwak, H.-B.; Min, K. Relationships between Socioeconomic Status, Handgrip Strength, and Non-Alcoholic Fatty Liver Disease in Middle-Aged Adults. International Journal of Environmental Research and Public Health 2021, 18, 1892. |
Reviewer 2 Report
I enjoyed reading your paper! Two minor suggestions: (1) you might mention that there are over 6,000 publicatio in PubMed on handgrip strength and health effects and (2) the x-axis on Figure 2 was hard to read so you might consider increasing the print size.
Author Response
Response to Reviewer 2 comments
We are very grateful for your interest and considerate comments from the reviewer to our study. As your recommendation, we have revised the paper. We hope that the reviewer agree to the revision.
1. You might mention that there are over 6,000 publication in PubMed on handgrip strength and health effects.
Answers) As your comment, handgrip strength have been used to be an attractive tool to stratify the risk of many diseases. Therefore, we have added the information that there are many articles on handgrip strength and its association with health problem in the introduction section.
|
Line 50, Introduction section) Furthermore, handgrip strength is an attractive tool to stratify the risk of many diseases. Therefore, there are many articles on handgrip strength and its association with health problem. Many studies have shown that handgrip strength is associated with metabolic disease, diabetes, and hypertension [13-15]. |
2. The x-axis on Figure 2 was hard to read so you might consider increasing the print size.
Answers) I agree to your suggestion. The characters under the x-axis were too small to read. Consequently, we have enlarged them. Because the figure can`t be uploaded to the clipboard, we have submitted the revised manuscript as a Word file.
Round 2
Reviewer 1 Report
Dear all,
I would like to thank you for responding to comments previously added to the manuscript. I would also like to thank you for the extensive work carried out in order to respond/add the information requested by me.
Like the authors, I have also focused on investigating the relationship between muscle strength (assessed by handgrip strength) and health outcomes (young, adult and elderly population). However, even though numerous studies have reported (the studies conducted by the authors, the studies conducted by myself - in addition to the studies summarized in the review studies) a direct relationship between handgrip strength and positive health outcomes, the "ideal" way of considering body size in determining muscle strength levels assessed by handgrip is not known (although numerous studies have addressed this issue). The provocation previously made by me regarding the need to test and report the inefficiency of the procedures available to normalize the values of handgrip strength is indeed necessary to inform the reader that although the results presented point to the use of grip strength manual as a diagnostic tool for health outcomes, the possible impact exerted by body size and its components (eg body fat) in determining the results identified in the association analyses, which may impact the direction of these findings cannot be disregarded.
Once again, congratulations for making these limitations explicit throughout the manuscript.
Reviewer 2 Report
I appreciate all the updates and I think your important paper is good for publication now and I have no additional suggested changes.